# Lipid Reshaping and Lipophagy Are Induced in a Modeled Ischemia-Reperfusion Injury of Blood Brain Barrier

**DOI:** 10.3390/ijms20153752

**Published:** 2019-07-31

**Authors:** Elena Lonati, Paola Antonia Corsetto, Gigliola Montorfano, Stefania Zava, Tatiana Carrozzini, Anna Brambilla, Laura Botto, Paola Palestini, Angela Maria Rizzo, Alessandra Bulbarelli

**Affiliations:** 1School of Medicine and Surgery, University of Milano-Bicocca, 20900 Monza, Italy; 2NeuroMi, Milan Center for Neuroscience, University of Milano-Bicocca, San Gerardo Hospital, 20900 Monza, Italy; 3Department of Pharmacological and Biomolecular Sciences, DiSFeB, Università degli Studi di Milano, 20133 Milan, Italy; 4Adienne Srl, 20867 Caponago, MB, Italy

**Keywords:** oxygen and glucose deprivation, lipophagy, lipid droplets, cholesterol, ischemia, vessel disease

## Abstract

Ischemic-reperfusion (I/R) injury induced a remodeling of protein and lipid homeostasis, under oxidative stress and inflammatory status. Starvation occurring during I/R is a condition leading to autophagy activation, which allows abnormal material clearance or amino acid, or both, and fatty acid (FA) recycling essential for survival. This study investigated the lipid reshaping, peroxidation, and related-signaling pathways, in rat brain endothelial cells (RBE4) subjected to 3 h of oxygen and glucose deprivation (OGD) and restoration of standard condition (I/R in vitro model). Lipids and proteins were analyzed after 1 or 24 h of oxygen and nutrient restoration. Together with the oxidative stress and inflammatory status, I/R injury induced a reshaping of neutral lipids and biogenesis of lipid droplets (LD) with excessive lipid storage. The increase of LC3-II/LC3-I ratio, an autophagy marker, and LC3 co-localization with LD suggest the activation of lipophagy machinery to counteract the cell engulfment. Lipophagy leads to cholesterol ester (CE) hydrolysis, increasing free cholesterol (FC) secretion, which occurred by specific transporters or unconventional exocytosis pathways, or both. Here, we propose that an unconventional spreading of FC and other lipid metabolites may influence the neurovascular unit (NVU) cells, contributing to Blood brain barrier (BBB) alteration or adaptation, or both, to the cumulative effects of several transient ischemia.

## 1. Introduction

The blood brain barrier (BBB) represents an important structure to regulate the exchanges between the peripheral circulation and the brain parenchyma [1]. According to BBB functional importance in controlling molecular traffic, the plasmalemma of brain endothelial cells is characterized by a specific membrane structure based on the unique combination of transport (efflux and influx transporters), metabolic (metabolic enzymes) and physical (lipid membrane with tight junctions) barrier properties [2]. Indeed, specific close-fitting membrane phospholipid (PL) and cholesterol domains [3] allow the flux of small gaseous molecules (oxygen, CO_2_) but not of certain hydrophobic molecules [4]. Thus, alterations in these properties lead to loss of BBB integrity and the consequent disruption of the neurovascular unit (NVU), a complex and well-organized cellular network (neurons, astrocytes, pericytes, brain endothelial cells, and extracellular matrix component) that physiologically preserves and sustains brain homeostasis [5].

The blood brain barrier (BBB) represents an important structure to regulate the exchanges between the peripheral circulation and the brain parenchyma [1]. According to BBB functional importance in controlling molecular traffic, the plasmalemma of brain endothelial cells is characterized by a specific membrane structure based on the unique combination of transport (efflux and influx transporters), metabolic (metabolic enzymes) and physical (lipid membrane with tight junctions) barrier properties [2]. Indeed, specific close-fitting membrane phospholipid (PL) and cholesterol domains [3] allow the flux of small gaseous molecules (oxygen, CO_2_) but not of certain hydrophobic molecules [4]. Thus, alterations in these properties lead to loss of BBB integrity and the consequent disruption of the neurovascular unit (NVU), a complex and well-organized cellular network (neurons, astrocytes, pericytes, brain endothelial cells, and extracellular matrix component) that physiologically preserves and sustains brain homeostasis [5].

In the last decade, several studies demonstrated that, ischemia-reperfusion (I/R) injury is one of the main causes in BBB break-down with the consequent enhancement in cerebral oxidative stress damage and neuroinflammatory response [6,7]. Alterations in lipid metabolism are strongly implicated in these events. Hypoxia, indeed, induces metabolic changes that influence fatty acid (FA) amounts and neutral lipid storage [8]. An overloading of lipid droplets (LD) might be potentially harmful since the gathering of neutral triglycerides (TGs) is accompanied by the accumulation of lipid species derived by the metabolism of the long chain FA and that are involved in lipoapoptosis [9,10]. Moreover, under oxidative stress, the increment of poly-unsaturated FAs (PUFAs) leads to their peroxidation, with the release of oxidative products, such as malondialdehyde (MDA) [11,12]. Kirac [13] and colleagues have recently shown that hepatic I/R injury induced lipid-mediated inflammatory endogenous pathways, altering levels of arachidonic acid (AA) and other omega-6 and omega-3 PUFAs.

Furthermore, our recent evidences demonstrated that, in brain endothelial cells subjected to oxygen and glucose deprivation and restoration (OGD/ogR), an in vitro modeling of I/R, the membrane cholesterol amount increased, while no changes in PLs and gangliosides (GG) were detected. The observed increase of cholesterol outcomes in the amyloidogenic metabolism and the amyloid-β (Aβ) production/accumulation during OGD/ogR [14].

All these events represent the additive stress generated in post-ischemic injury and are strictly implicated in the progressive decline of cognitive functions. The present study is a step forward in the evaluation of lipid alterations under OGD/ogR treatment, by monitoring fluctuations in FA and neutral lipid levels, as well as pathways linked to them.

## 2. Results and Discussion

### 2.1. OGD/ogR Rearranges Rat Brain Endothelial Cells (RBE4) Total FA Profile

Alteration in level and metabolism of lipids, as well as their localization in cellular membranes, is strictly related to cell dysfunctions. Under ischemia-reperfusion, lipid changes are targets and players of oxidative stress and inflammation that mediate death or survivor messages. For these reasons lipid fluctuations may represent an important mechanism involved in endothelial cell reaction to ischemic injury. Therefore, we evaluated the possible modification in endothelial cell lipid composition induced by OGD/ogR; firstly, we analyzed the total FA composition in RBE4 after 1 and 24 h of ogR. The complete FA cell profile is reported in Appendix A; after treatment, it is possible to observe a significant increase of all PUFAs, in particular at ogR1h compared to control.

The relevant results are depicted in Figure 1; considering whole cell FA profile, the major FAs modified were the n-6 AA (AA C20:4) and n-3 PUFAs, such as eicosapentaenoic acid (EPA, C20:5) and docosahexaenoic acid (DHA, C22:6) (Figure 1A), that significantly increase at ogR1h, remaining higher than control at ogR24h.

To understand if FA composition changes were reflected in the FA profile of a specific lipid class, we purified and analyzed also the neutral and PL FA profiles.

After HPLC purification of neutral lipids, their FA percentage distribution showed no substantial change of total PUFA at ogR1h followed by a significant decrease during the successive 24 h (Table 1). Nevertheless, analyzing the two classes of PUFAs, we observed a different trend: n-6 PUFA, in particular AA, significantly decreased during ogR (24 h), while n-3 PUFA were significantly enriched at ogR1h, returning to control levels at ogR24h (Table 1).

On the other hand, the analyses of the different PL classes showed no changes in FA composition except for the phosphatidylcholine (PC) that was significantly enriched in AA and in EPA (Figure 1B). It is worth of note that the increase in AA in PC corresponds to a decrease in neutral lipids, suggesting a remodeling of this specific FA through the lipid species. The increment in membrane resident-PUFAs, AA in particular, during reoxygenation may represent a damaging event for cells, since they are the main substrates for lipid peroxidation under oxidative conditions. In parallel, n-3 PUFA enrichment in neutral lipids (TG and CE) at ogR1h might be a mechanism for FA storage in LD to support the cell lipid homeostasis [15,16]. In order to deeply understand these pathways, we evaluated lipid peroxidation products and AA cascade on one side, as well as TG amount and LD formation on the other side.

### 2.2. Lipid Peroxidation and AA Cascade Activation Under OGD/ogR

Accordingly to previously literature [13,17], we evaluated the proteins involved in AA metabolism. In particular, we analyzed protein levels of cPLA2, the key enzyme in AA release from PC [18], and of inducible COX-2 that catalyzes the conversion of AA in inflammatory mediators (prostaglandins, thromboxane, leukotrienes). Recent findings have already highlighted the activation of the cPLA2/AA/COX-2 pathway following cerebral [17] or hepatic ischemic injury [13] as an inflammatory response.

Therefore, RBE4 cells, subjected to OGD treatment for 3 h, were successively harvested at 1 and 24 h of ogR to evaluate cPLA2 in membrane-enriched fractions (MEF), while COX-2 was evaluated in the whole cell homogenate.

As shown in Figure 2A, cPLA2 protein level increased at ogR1h of about 60%, remaining higher than the control at ogR24h. In the same way, we observed a significant 90% increase of COX-2 protein at ogR1h that enhanced up to 270% at ogR24h (Figure 2B).

COX-2 and cPLA2 increased together with the AA increment trend in PL and in the whole cell are in line with the literature evidences on cPLA2/AA/COX-2 pathway activation during the 24 h following reperfusion in an in vivo model of transient focal cerebral ischemia (tFCI) [17]. Interestingly, our data suggest that the pathway of AA refurbishment in PL might be mediated by a decrease of AA storage in neutral lipids and that this lipid reshaping could be at the base of the inflammatory response.

In parallel, MDA was evaluated as PUFA peroxidation product. MDA content increased in a time dependent-manner along ogR in endothelial cells, suggesting a persistent condition of oxidative stress (Figure 3).

### 2.3. OGD/ogR Induces TG Increase and LD Formation

Hypoxia has been shown to stimulate lipid storage and inhibit lipid catabolism in cultured cardiac myocytes and macrophages [8], although the effects of re-oxygenation are not well understood from this point of view. Interestingly, under our experimental conditions, we observed a significant time dependent increment in TG content along to ogR (Figure 4). TG increase might be due to the activation of hypoxia-inducible factor 1α (HIF-1α), which seems to regulate lipin-1 [15], a protein that catalyzes the penultimate step of TG synthesis [19]. Accordingly, under our experimental conditions, we observed the increase of HIF-1α [20].

Moreover, suppression of FA oxidation and increased lipid storage capacity was associated with hypoxia inducible factors activation [15,21]. Therefore, we hypothesize that newly synthetized TG might be quickly stored in LD.

To investigate this hypothesis, LD were marked with BODIPY 493/503 specific for neutral lipids. The results are shown in Figure 5, where it is possible to observe LD increase after 1 and 24 h ogR.

In particular, cells were stained also with DAPI and phalloidin to mark cell shape and cytoskeleton. Notably, lipid bodies appear localized in perinuclear area in polarized shape at ogR1h (Figure 5B), while, after 24 h ogR, they seemed to move towards plasma membrane following the specific direction of actin filaments (Figure 5C).

These data suggest that de-novo generated LD might be committed to exocytosis after 24 h from restoration. Indeed, inflammatory conditions increase BBB transcytosis process that is responsible of different macromolecule uptake or release, or both, to neuronal cells [22]. n-3 PUFAs, which are not synthetized by neurons, must be delivered to brain parenchyma where they can exert their neuroprotective effects against ischemic insult [23,24].

### 2.4. OGD/ogR-Induced Lipophagy

Oxygen and nutrient starvation leads to LD biogenesis with the specific aim to maintain the redox homeostasis and managing the lipid unbalance when microenvironment rapidly changes, as in the case of ogR [25]. Interestingly, both autophagy and cPLA2 protein seem to play a role in LD generation, mobilizing the available endogenous and structural lipids, in a fine equilibrium between FA-induced lipotoxicity and FA employment for cell survival [26,27].

Autophagy is a catabolic process that is activated under diverse stressful conditions, including OGD [28], to prevent accumulation of damaged cellular components (proteins, lipids, mitochondria, etc.). Indeed, nutrient deprivation, hypoxia, elevated ROS levels, lipid overload, and increased unsaturated FAs are all events that occur in OGD and that lead to autophagy induction [25]. Moreover, autophagy flux is affected also by the presence of intracellular MDA, triggering an atypical autophagy pathway independent from Beclin-1 [29], a protein involved in the regulation of initiation process of macroautophagy. Macroautophagy is characterized by the biogenesis of autophagy vacuoles (AV), marked by the PE-conjugated form of microtubule-associated proteins 1A/1B light chain 3B (LC3-II). Low levels of MDA, indeed, increased LC3-II/LC3-I ratio (lapidated/de-lipidated form) and decreased p62 levels (an autophagy-selective substrate), while high levels seems to inhibit the flux [29].

Thus, considering the observed LD biogenesis and the increase of MDA along the ogR, we evaluated Beclin-1, LC3II/I ratio and p62. As observed by Ye and colleagues [29], Beclin-1 levels did not change after OGD/ogR, while LC3II/I ratio significantly increased at ogR1h (+80%), decreasing in the following 24 h (+32% vs. CTRL). A time-dependent p62 decrease was observed during the ogR, statistically significant at 24 h (Figure 6). Data, hence, suggest the activation of such autophagy machinery with AV biogenesis occurrence.

Under starvation conditions, LC3-II seems to co-localize with LD [30,31], indicating a fusion between LD and AV with the generation of lipo-autophagosomes, and so the occurrence of lipophagy (Figure 7). This autophagy pattern is activated in order to deliver lipids to lysosomes or to exocytosis. Accordingly, immunofluorescence analysis demonstrated a co-localization between LD (green) and LC3 (red) at ogR1h (Figure 7K) that seems to reduce at ogR24h, although a high signal was detected (Figure 7L).

Altogether, our data suggested that the autophagy machinery might regulate both AV and LD biogenesis as a coordinated mechanism for the maintenance of lipid cellular homeostasis.

### 2.5. OGD/ogR-Induced CE Hydrolysis and FC Release in Extracellular Milieu

We tried to disclose whether LD during ogR were committed to lysosomal degradation or exocytosis, or both, taking in consideration possible changes in cholesterol ester (CE) levels, another lipid which is accumulated in lipid cellular bodies together with TG as a source of FAs. Remarkably, we observed in treated cells that CE fraction at ogR1h significantly decreased about 40%, from 13.04 to 8.12 µg/mg proteins, while at ogR24h, CE content returned to the control condition (Figure 8A); this data might indicate a possible CE hydrolysis by lysosomal mechanism.

CE decrease might play a role in the previously described free cholesterol (FC) enrichment at ogR1h in cellular membranes [14], probably influencing the FC availability. Notably, the increase of CE hydrolysis was discovered along with Aβ stabilization and enhancement [32], which we also observed in our ischemic model [20].

Thus, we got an overall picture of cholesterol distribution in cells during the treatment, and we observed a consistent alteration compared to the control (26% CE and 74% FC, molar percentage) at ogR1h (17% CE and 83% FC), as shown in Figure 8B. FC availability under OGD/ogR could be a damaging event that needs to be counteracted with its efflux from BBB endothelial cells. Indeed, hydrolysis of LD-associated CE is recognized as a fundamental step for cholesterol efflux [33] that is associated to autophagy machinery activation in lipid-loaded cells [34,35]. Therefore, we investigated the amount of FC released in the extracellular medium, finding an increase of about 75% at ogR1h, from 3,75 µg/mg proteins to 6,62 µg/mg proteins. Then, extracellular FC amount did not significantly change during the following 24 h of recovery with respect to the first h (Figure 8C). The way in which FC is released outside cells need to be elucidated. Here, we propose that the FC efflux occurred by specific active transporters, such as ATP binding cassette (ABC) transporters, for lipophilic and amphipathic molecules, including lipids and xenobiotics. Since data from the literature indicate that ischemia might influence the expression of different ABC pumps [36], we analyzed the protein levels of drug transporter ABCB1/MDR1. Data showed a time-dependent increase in MDR1 during ogR, with a significant increment at ogR24h, suggesting a possible ABC transporter involvement in extracellular FC release (Figure 9).

Interestingly, the increased protein level of cholesterol transporters has been also associated to release of extracellular vesicles (EVs) [37]. Several evidences described the role of EVs in ensuring cholesterol homeostasis and highlighted the presence of cholesterol-rich lipid rafts region in EVs membrane [37]. Thus, FC export might also occur by fusion of LD with other organelles releasing EVs [38], or by direct fusion with plasma membrane through an unconventional secretory pathway driven by the chaperone p62 carrier [9]. Indeed, as shown in Figure 6, p62 protein levels decrease at ogR24h when LD seems to be addressed to secretion, indicating that several mechanisms might be involved in cholesterol homeostasis during reperfusion.

## 3. Materials and Methods

### 3.1. Materials

All commercial chemicals were of the highest available grade. Solvent solutions were from Carlo Erba (Milano, Italy). All powdered reactants, solutions for electrophoresis, Phalloidin Tetramethylrhodamine B isothiocyanate, DAPI, and anti-β-actin antibody, were from Sigma Chemical Co (St. Louis, MO, USA). Lipid standards, FA s, neutral lipids, PL, and Avanti Polar Lipids were from Sigma Chemical Co. (St. Louis, MO, USA). The 5% CO_2_:95% N_2_ gas cylinder came from Sapio (Monza, Italy). Collagen I rat tail solution for RBE4 cell culture, BODIPY 493/503, and Alexa Fluor^®^ 567-labeled goat anti-rabbit IgG were from Invitrogen, Life Technologies Italia Fil. (Monza, MB, Italy). Stock solutions for RBE4 cell culture included: alpha-MEM medium with glutamax-1 and Ham’s F-10 nutrient medium, geneticin solution antibiotic, and Euroclone (Milano, MI, Italy). The complete protease inhibitor cocktail came from Roche Diagnostics S.p.A (Milano, Italy). Anti-cPLA2 antibody came from Santa Cruz Biotechnology Inc. (Santa Cruz, CA, USA). Anti-Cox-2, anti-LC3-I/II anti-p62, and anti-Beclin-1 antibodies were from Cell Signaling Technology (Danvers, MA, USA). Secondary horseradish peroxidase (HRP) -conjugated antibodies and enhanced chemiluminescence (ECL) SuperSignal detection kit were from Pierce (Rockford, IL, USA).

### 3.2. Cell Cultures

The rat brain endothelial (RBE4) clone showed typical endothelial morphology and retains many brain endothelial cell characteristics [39,40,41,42,43]. RBE4 cells, provided as a gift by M. Aschner (Department of Pediatrics, Vanderbilt Kennedy Centre, Nashville, TN, USA), were plated on collagen (50 µg/mL in acetic acid 0.02 M)-coated dishes or flasks and were grown in the presence of 44% alpha-MEM: 44% F-10 Nutrient supplemented with 10% heat inactivated fetal bovine serum (FBS). RBE4 cells were plated in collagen-coated dishes (12,500 cells/cm^2^) and maintained at 37 °C in a 5% CO_2_ atmosphere for three days (80% confluence) before treatment.

### 3.3. OGD Treatment

RBE4 cells were subjected to OGD for 3 h, as previously described [14,20]. Afterward, OGD, normoxic and normoglycemic conditions were restored for 1 h and 24 h. Cells were replaced in normal culture conditions (37 °C in a 5% CO_2_ atmosphere), and BSS in each dish was supplemented with restoration solution containing 5mM glucose and 10% FBS in culture medium. From here on, we will use the term “oxygen and glucose Restoration” (ogR) to indicate “normoxic and normoglycemic conditions restoration post-OGD”. RBE4 untreated cells were maintained in normal culture conditions and collected together with treated cells subjected to ogR for 24 h.

### 3.4. Homogenate Obtainment and MEFs Isolation

At 1 and 24 h of ogR, cells were scraped in PBS containing the protease inhibitor cocktail. Pellets were homogenized with hypotonic solution (Tris-HCl pH 7.4 1 mM, EDTA pH 7.4 1 mM, KCl 15 mM, NaCl 30 mM, protease inhibitor cocktail), then an equal volume of isotonic solution 2X (sucrose 0.5M, Tris-HCl pH 7.4 2 mM, EDTA pH 7.4 2 mM, protease inhibitor cocktail) was added. To isolate MEFs, samples were centrifuged 12 min at 800× *g* in order to separate nuclei-enriched pellet and post nuclear supernatant (PNS). PNS was ultracentrifuged 1 h at 100,000× *g*, and the resultant membrane-enriched pellet was resuspended in an MBST buffer (MES pH 6.5 25 mM, NaCl 150 mM, Triton X-100 1%, PMSF 1 mM, protease inhibitor cocktail). cPLA2 protein levels were analyzed in MEFs.

### 3.5. Electrophoresis and Immunoblotting

Protein analysis was performed by SDS-PAGE electrophoresis on 12% polyacrylamide tris-glycine, protein transfer to a nitrocellulose membrane (Amersham, GE Healthcare Europe GmbH, Milano, Italy), revelation by Ponceau staining (Sigma Chemical Co., Milano, Italy), and immunoblotting with specific antibodies. Electrophoresis separation and western blotting were carried out on equal amounts (as protein) of homogenate or MEF samples, in order to investigate protein level expression in cellular membranes. Nitrocellulose membranes were blocked in TBS-Tween 0.1% or 0.2% buffer containing 5% non-fat milk or 3–5% BSA and probed with specific antibodies diluted in the same solution. Immunoblottings were performed using anti-cPLA2 (1:200), anti-COX2 (1:1000), anti-LC3 I-II (1:1000), anti-p62 (1:1000), anti Beclin-1 (1:1000), and anti-β-actin (1:1500) antibodies. Immunoreactive proteins were revealed by ECL and semi-quantitatively estimated by LAS4000 Image Station (GE Healthcare Italia, Milano, MI, Italy). Normalization in the same sample was carried out with respect to β-actin homogenate samples [44,45] and to the total amount of proteins detected by Ponceau staining in MEFs, permitting a straightforward correction for lane-to-lane variation [46,47].

### 3.6. Lipid Extraction and Quantification

Cell and cell medium lipids were extracted according to Folch, with minor modifications, as previously described [48]. Briefly, cell pellets were homogenized with chloroform/methanol 1:2, centrifuged to recover the lipid extract and extracted again two times. Cell medium was lyophilized before extraction. Each solvent used for extraction and analysis contained 0.045 mM 3,5-di-tert-4-butylhydroxytoluene (BHT) to avoid PUFA oxidation.

Purification of single PL moieties was achieved with an High Performance Liquid Chromatography-Evaporative light scattering detector (HPLC-ELSD) system equipped with a silica normal-phase LiChrospher Si 60 column (length 250 mm, I.D 4.6 mm, and film thickness 5 μm, Merck, Darmstadt, Germany). The chromatographic separation was achieved as previously described [49]. For PL quantitation, a mixture of standard PL was injected prior to samples in three different concentrations to construct a calibration curve for each single PL species [50].

Neutral lipids from cellular and medium lipid extract were separated and accurately quantified by high performance liquid chromatography on a LiChrospher SI 60 5 mm, 250 × 4 HPLC column, and using ELSD detector according to [51]. The mobile phase consisted of solvent A: isooctane, tetrahydrofuran 99:1 *voL*/*voL* and solvent B: acetone, dichloromethane 2:1 *voL*/*voL*. The flow rate was 1.6 mL/min. ELSD detector was settled as follows: pressure 3.5 bar, temperature 50 °C, gain 8. The injection volume was 20μL. Four different concentrations of free and esterified cholesterol and TGs standards were injected prior to sample to construct a calibration curve.

The total FA composition of samples and purified phospho- and neutral-lipids was determined by gas chromatography (Agilent Technologies 6850 series II, Santa Clara, CA, USA) as previously described [52]. Lipids were derivatized (sodium methoxide in methanol 3.33% (*w*/*v*)) to obtain FA methyl esters (FAME). Prior to derivatization, a known amount of internal standard (IS) (C17:0 TG) was added to each sample to correct for yield and recovery of the reaction. A standard mixture containing all FAME was injected as calibration for quantitative analysis.

### 3.7. MDA Analysis

Lipid peroxidation was assessed measuring MDA, according to Karatas and colleagues [53], with slight modifications. Malondialdehyde was determined by HPLC (Jasco, Japan) equipped with a UV detector. A C18 column was used at ambient temperature. MDA standards were prepared from TEP (1,1,3,3-tetraethoxypropane) as previously described [49]. For sample preparation, cell lysates were treated with 0.1 M HClO4 to allow protein precipitation and release of bound MDA. Samples were centrifuged at 4500g for 5 min and supernatants were used for HPLC analysis. The mobile phase was 30 mM KH_2_PO_4_/methanol/acetonitrile (72/18/11, *v*/*v*); the flow rate was 1 mL/min. Chromatograms were monitored at 254 nm.

### 3.8. LD and LC3 Immunostaining

At 1 and 24 h of OgR, RBE4 cells, grown on glass coverslips, were fixed in 4% paraformaldehyde at room temperature for 20 min and permeabilized at room temperature for 6 min with 0.5% triton X-100 in Hepes Buffer (4-(2-hydroxyethyl)-1-piperazineethanesulfonic acid) 20 mM. To stain actin filaments, cells were incubated with phalloidin-TRITC dilute 1:40 in PBS, at room temperature, for 40 min. To stain AVs, cells were incubated with LC3A/B (D3U4C) XP^®^ Rabbit antibody, 1:100, at room temperature, for 120 min and, subsequently, incubated with secondary antibody Alexa Fluor^®^ 567-labeled goat anti-rabbit IgG, 1:1000, at room temperature, for 30 min, then cells were washed five times with PBS and incubated with BODIPY 493/503 1:100, at room temperature, for 15 min. Nuclei were stained with 4′,6-diamidino-2-phenylindole (DAPI)1:1000, at room temperature for 5 min. Fluorescence images were collected under a Nikon Eclypse TE200 inverted microscope with immersion objective at 60× magnification and photographed with Nikon digital camera (Nikon, Japan).

### 3.9. Statistical Analysis

All data are expressed as mean ± S.E. of three separate experiments performed in triplicate. The differences were calculated by means Student’s t test. A *p* value < 0.05 was considered to be statistically significant with respect to the control and is indicated by an asterisk, *p* value < 0.01 is indicated by two asterisks, and *p* value < 0.001 is indicated by three asterisks. § indicates a *p* value < 0.05 with respect to ogR1h.

## 4. Conclusions

In this study we revealed a reshaping in the lipid profile of RBE4 cells in response to stress conditions induced by ischemic injury. In particular, we found AA enrichment in PC, probably refurbished from neutral lipids, in parallel to the increment of lipid peroxidation products and the activation of the cPLA2/AA/COX-2 cascade. Accordingly, lipophagy activation at ogR1h occurs to effectively remove lipid excess modulating lipid homeostasis and intra- extra- cellular signaling. LD could be driven, at least in part, to unconventional exocytosis, as suggested by immunofluorescence data of ogR24h, probably to counteract the intracellular TG overload [9]. In addition, the secretion of n-3 PUFA contained in LD might positively influence neighboring cells of NVU, promoting an anti-inflammatory phenotype of microglia [54]. Indeed, PUFA exchange from endothelial cells to neurons has been associated to neurotransmission and neuroprotection [55]. On the other hand, exocytosis of lipid metabolites (i.e., ceramides, diacylglycerols, and PUFA derived lipid mediators, obtained by TG in LD) might induce pro-apoptotic signals in the adjacent cells [10].

Moreover, lipophagy induced to prevent intracellular cholesterol accumulation could become a doubled-edged sword since, according to our previous evidences [14], FC released by hydrolysis of CE affects the cell membrane composition leading to Aβ production. Furthermore, cholesterol seems to play a role in the regulation of EVs release, enhancing the cholesterol-rich microparticle (MP) shedding [37]. In fact, elevated levels of circulating MP in plasma have been associated to several vascular disease, including acute ischemic stroke [56]. How cell to cell communication, occurring in response to I/R injury, may negatively or positively influence the NVU is a key point of analysis, in particular in those pathological conditions of repeated ischemia manifesting the cumulative effects of several transient ischemic injuries.

## Figures and Tables

**Figure 1 ijms-20-03752-f001:**
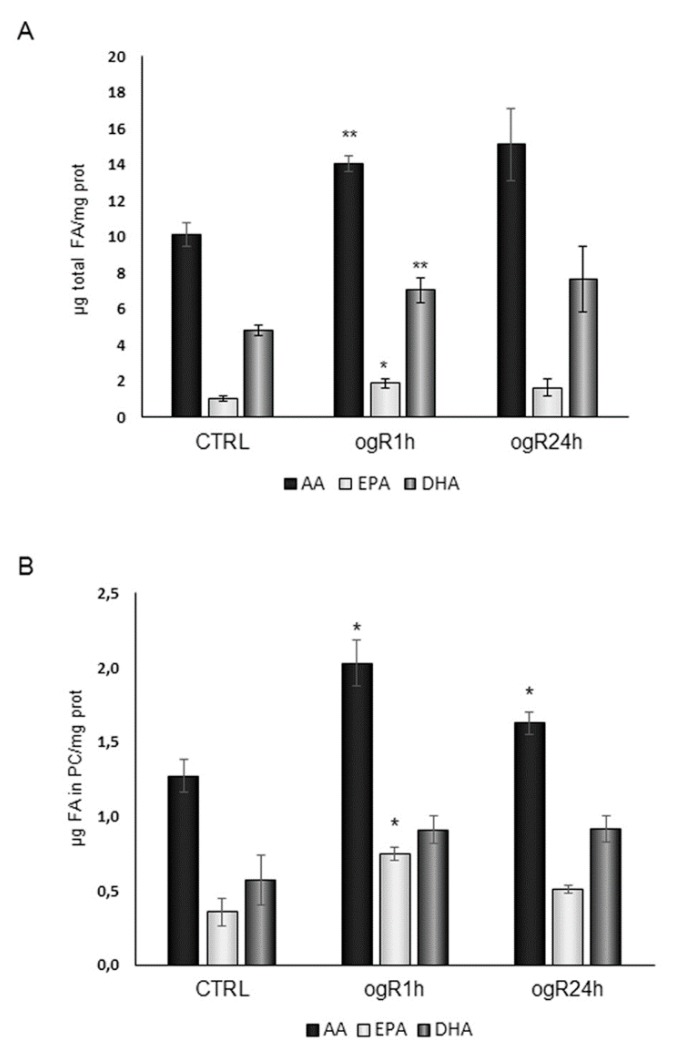
Total polyunsaturated fatty acid (PUFA) content of rat brain endothelial cells (RBE4) cells and of purified phosphatidylcholine (PC) after oxygen and glucose deprivation and restoration (OGD/OgR) treatment for 1 and 24 h. Cells, after OGD/ogR treatment, were collected and lipids were extracted by different mixture of methanol/chloroform. Cell FAs (fatty acids) were measured as FA methyl esters (FAME) by a GC (Agilent Technologies 6850 series II, Santa Clara, CA, US) equipped with flame ionization detector. FAME were quantified using the chromatographic peak area according to the internal standard (IS) method. Histograms represent the most relevant results of whole cell (**A**), and PC (**B**) FA composition. The data are normalized to cellular protein content and expressed as mean ± S.E. from three independent experiments, * *p* < 0.05, ** *p* < 0.01 vs. control.

**Figure 2 ijms-20-03752-f002:**
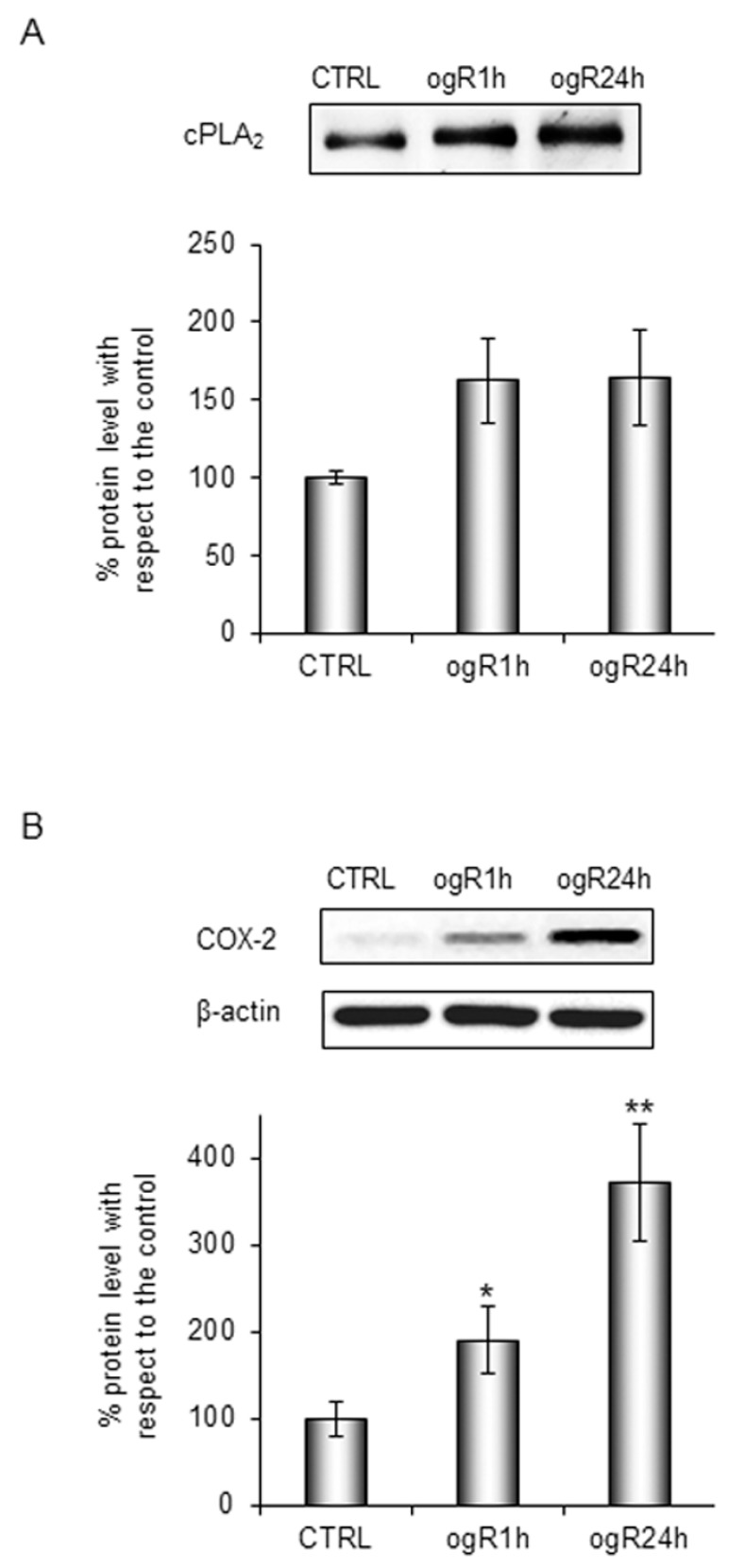
Analysis of proteins involved in arachidonic acid (AA) metabolism in ogR. Cells subjected to OGD treatment were harvested in hypotonic solution at 1 h and 24 h ogR. Total homogenate aliquot was collected and membrane-enriched fractions (MEFs) were obtained by ultracentrifugation. Equal amounts of MEF or homogenate samples (as protein) were subjected to SDS-PAGE and WB analysis. Ponceau staining of total lanes was employed to perform band quantification of MEF samples, while total lysates were normalized by β-actin content. Panel (**A**) represents the % of cPLA2 increment and panel (**B**) the % of COX-2 protein levels with respect to CTRL. The data are expressed as mean ± S.E. from three independent experiments, * *p* < 0.05, ** *p* < 0.01 vs. control.

**Figure 3 ijms-20-03752-f003:**
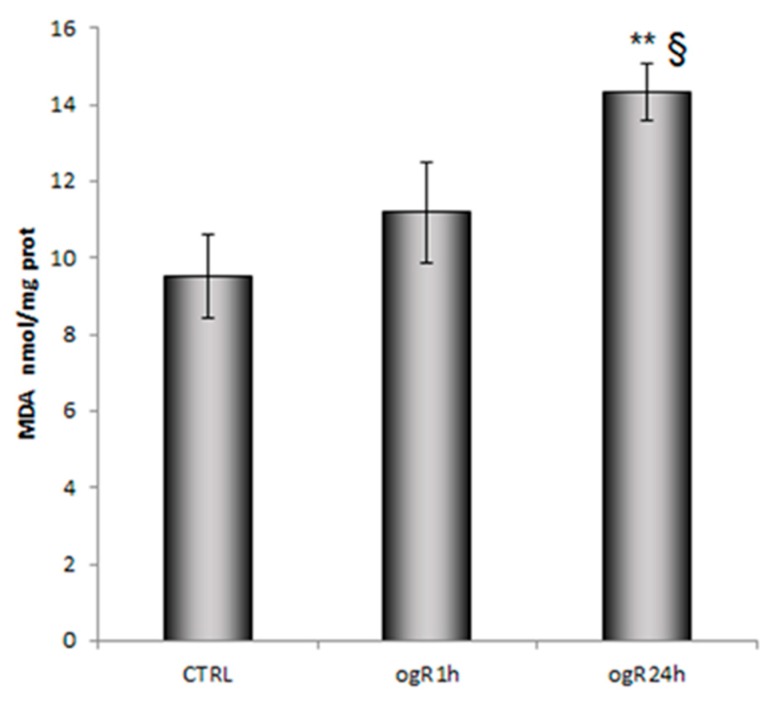
Evaluation of lipid peroxidation in RBE4 after OGD/ogR. Lipid peroxidation was evaluated by measuring malondialdehyde (MDA) content in control and OGD/ogR RBE4 cells as marker of lipid peroxidation by HPLC- Evaporative light scattering detector (ELSD) system. The data are normalized to cellular protein content (nmol/mg protein) and expressed as mean ± S.E from three independent experiments, ** *p* < 0.01 vs. control, § *p* < 0.05 vs. ogR1h.

**Figure 4 ijms-20-03752-f004:**
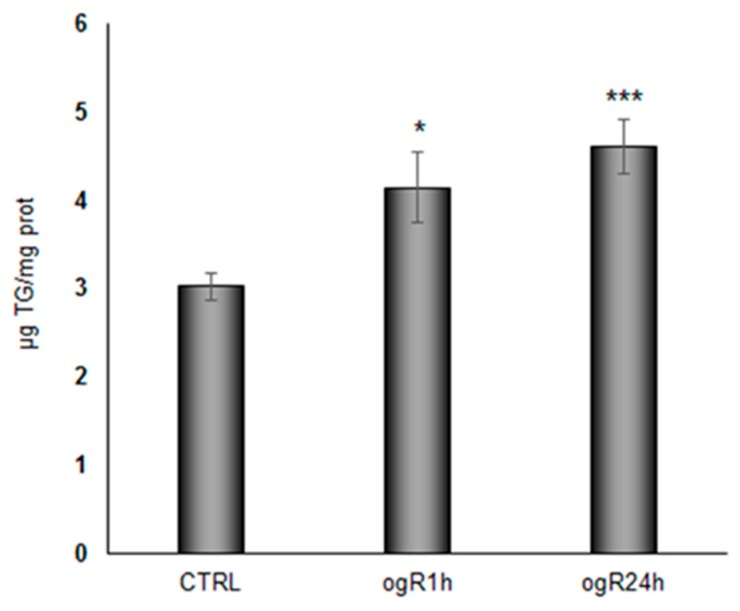
Triglyceride (TG) content in RBE4 after OGD/ogR. Cells, after OGD/ogR treatment, were collected and lipids were extracted by different methanol/chloroform mixtures. TGs were quantified by HPLC-ELSD system. Histogram represents the mean ± S.E. of TG concentration normalized to protein content in the same sample. * *p* < 0.05, *** *p* < 0.001 vs. control, *n* = 3.

**Figure 5 ijms-20-03752-f005:**
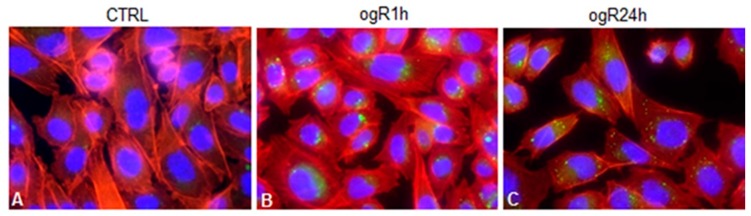
Lipid droplet (LD) staining in RBE4 after OGD/ogR. RBE4 cells were fixed in 4% paraformaldehyde and permeabilized at 4 °C with 0.5% Triton X-100 Hepes Buffer (4-(2-hydroxyethyl)-1-piperazineethanesulfonic acid. Images show LDs stained with BODIPY 493/503, actin filaments stained with phalloidin and nuclei stained with DAPI (blue fluorescence) in control (**A**), ogR1h (**B**), and ogR24h (**C**) RBE4 cells. Fluorescence images were collected under a Nikon Eclypse microscope with immersion objective at 60× magnification.

**Figure 6 ijms-20-03752-f006:**
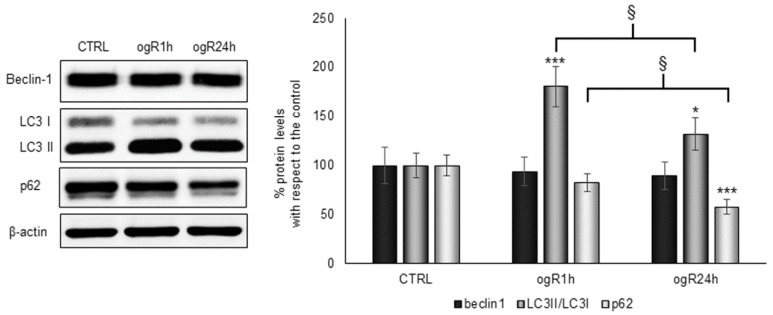
Autophagy marker analysis in RBE4 after OGD/ogR. Cells subjected to OGD/ogR treatment were harvested in lysis buffer, then equal amounts of homogenate samples (as protein) were analyzed by WB. LC3II/I, p62 and Beclin-1 were detected with specific antibodies and revealed by enhanced chemiluminescence (ECL). Sample were normalized on β-actin immunoreactivity. Histograms represent the % of protein levels respect to control as mean ± S.E. from three independent experiments, * *p* < 0.05, *** *p* < 0.001 vs. control; § *p* < 0.05 vs. ogR1h.

**Figure 7 ijms-20-03752-f007:**
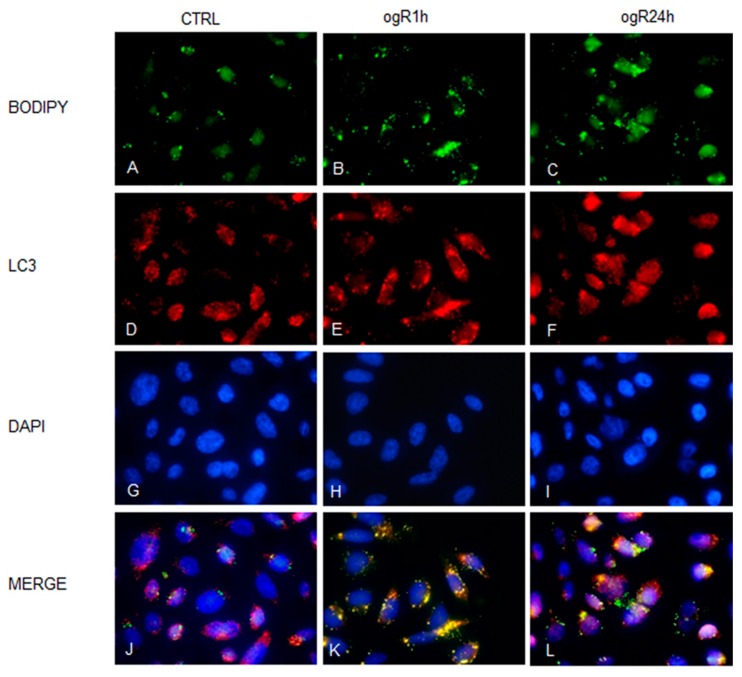
LD increase under OGD/ogR. RBE4 cells were fixed in 4% paraformaldehyde and permeabilized at 4 °C with 0.5% Triton X-100 Hepes Buffer. Then cells were tripled marked with LC3 specific antibody/and secondary antibody Alexa Fluor^®^ 567-labeled (red fluorescence) to stain autophagy vacuoles (AVs), with BODIPY 493/503 (green fluorescence) to stain LD, and with DAPI for nuclei staining in control (**A**,**D**,**G**,**J**), ogR1h (**B**,**E**,**H**,**K**) and ogR24h (**C**,**F**,**I**,**L**) RBE4 cells. Fluorescence images were collected under a Nikon Eclypse microscope with immersion objective at 60× magnification.

**Figure 8 ijms-20-03752-f008:**
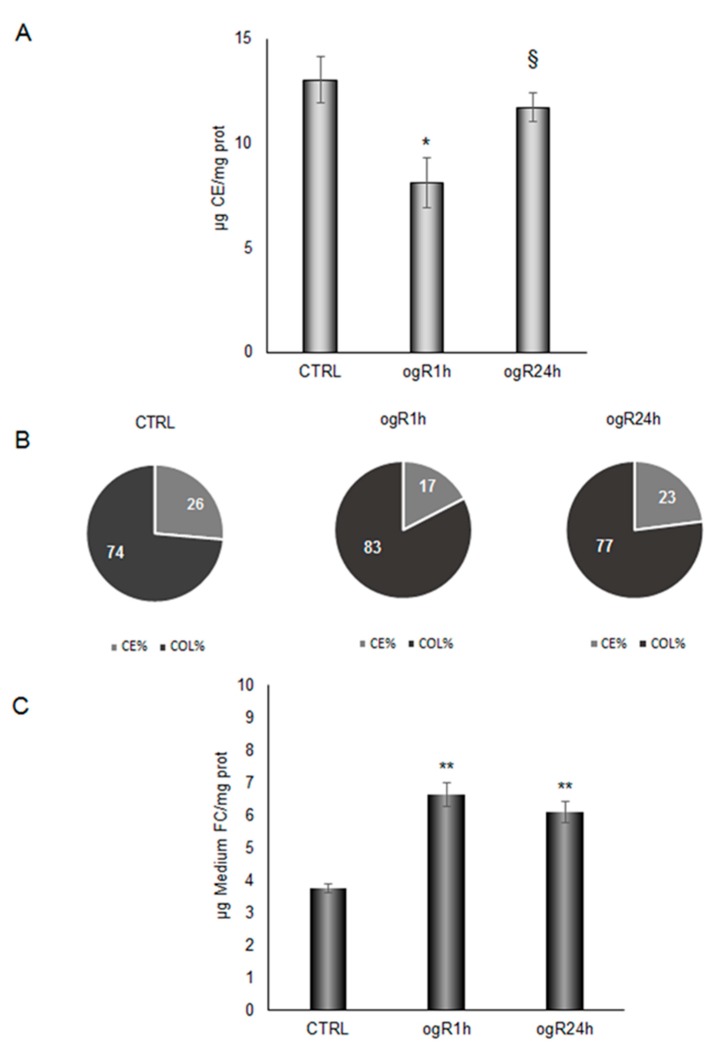
Changes in intracellular and extracellular cholesterol content under OGD/ogR. Cells, after OGD/ogR treatment, were collected and lipids were extracted by different mixture of methanol/chloroform. Free cholesterol (FC) and cholesterol ester (CE) were evaluated by HPLC-ELSD. Panel A represents intracellular CE concentration normalized to protein content. Panel (**B**) depicts with circle charts the molar percentage distribution of intracellular CE and FC. Panel (**C**) reports the cell medium FC content normalized for protein of cells seeded in each analyzed plate. Data are reported as mean ± S.E. (**A**,**C**) from three independent experiments, * *p* < 0.05, ** *p* < 0.01 vs. control; § *p* < 0.05 vs. ogR1h.

**Figure 9 ijms-20-03752-f009:**
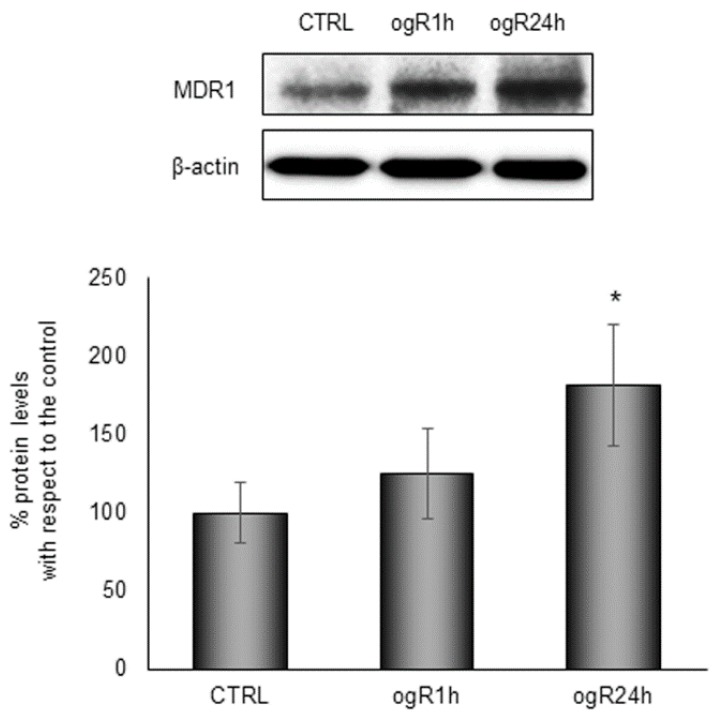
Evaluation of MDR1 protein levels inRBE4 after OGD/ogR. Cells subjected to OGD treatment were harvested in lysis buffer, then equal amounts of homogenate samples (as protein) were subjected to SDS-PAGE and WB analysis. MDR1 was detected with the specific antibody and revealed by ECL. Sample were normalized on β-actin immunoreactivity. Histograms represent the % of protein levels respect to control as mean ± S.E from three independent experiments, * *p* < 0.05 vs. control.

**Table 1 ijms-20-03752-t001:** FA composition of neutral lipids purified by High Performance Liquid Chromatography (HPLC) from RBE4 endothelial cells after OGD/ogR treatment for 1 and 24 h.

	%					
	CTRL		ogR1h		ogR24h	
	Mean ± S.E.		Mean ± S.E.		Mean ± S.E.	
C16:0	19.308	1.025	18.683	0.634	18.820	0.098
C16:1	6.219	0.474	6.288	0.600	5.579	0.720
C18:0	11.560	1.490	9.620	0.432	10.354	1.141
C18:1	40.927	2,685	43.309	0.981	48.557	1.557
C18:2	9.002	1.297	7.996	0.448	5.982	0.379
C18:3 n-3	0.942	0.181	1.102	0.252	1.265	0.118
C20:3	1.173	0.406	0.674	0.385	0.593	0.315
C20:4 n-6	4.294	0.066	3.962	0.127	**3.217**	**0.107**
C20:5	4,425	0.621	5.386	0.319	3.794	0.756
C22:5	1.147	0.040	1.774	0.356	**0.625**	**0.083**
C22:6	1.002	0.140	1.206	0.069	1.214	0.267
PUFA	21.985	1.414	22.099	1.143	**16.690**	**0.848**
n-6 PUFA	14.469	1.425	12.631	0.854	**9.791**	**0.647**
n-3 PUFA	6.979	0.342	**9.467**	**0.387**	6.899	0.965
Total FA (µg/mg prot)	20.634	2.497	16.929	0.998	23.904	0.699

Data are reported as mean ± S.E. from three independent experiments, bold numbers indicate a significant statistical difference vs. control (*p* < 0.05).

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
