# Peer review of "Lipid Reshaping and Lipophagy Are Induced in a Modeled Ischemia-Reperfusion Injury of Blood Brain Barrier"

_ijms, 2019, doi:10.3390/ijms20153752_

Round 1

Reviewer 1 Report

The manuscript by Lonati and colleagues aims to show that the effects of oxygen deprivation on the integrity of rat brain endothelial cells. Overall the manuscript is well written with a well-designed study. I believe the manuscript is acceptable for publication after the authors make minor edits.

The figures for the manuscript are of poor quality. The y-axis labels need to be more consistent across figures. For example, in figure 1A the axis ranges from 0-20 and then in 1B the axis ranges from 0,0 to 2,5. Is this correct?

What do the asterisks denote in the figures? 

Moreover, none of the figures have figure legends. Please add figure legends for all figures. 

Author Response

Point 1.

The figures for the manuscript are of poor quality. The y-axis labels need to be more consistent across figures. For example, in figure 1A the axis ranges from 0-20 and then in 1B the axis ranges from 0,0 to 2,5. Is this correct?

Thank you for suggestions. We tried to improve the readability of the images in the text.

About the question, considering that the content on fatty acids in phosphatidylcholine is much lower compared to total cell fatty acid content, it is necessary represent data with different axis ranges.

Point 2-3.

What do the asterisks denote in the figures? Moreover, none of the figures have figure legends. Please add figure legends for all figures.

We apologize for it. There was an editorial problem during the upload of the paper and the figure legends were erroneously inserted in the text. Now, we have moved all the figure legends immediately below the figure, changing the size character from 13px to 12px. The meaning of the asterisks is stated at the end of the figure legends.

Figure 1, page 3 lanes 87-95

Figure 2, page 5 lanes 132-138

Figure 3, page 6 lanes 151-155

Figure 4, page 7 lanes 171-174

Figure 5, page 7 lanes 184-189

Figure 6, page 8 lanes 221-226

Figure 7, page 9 lanes 236-241

Figure 8, page 10 lanes 254-261

Figure 9, page 11 lanes 285-290

Reviewer 2 Report

Figure 1 is a little confusing. In lines 78-80 it mentions that there is a significant increase in PUFAs at 1 hr and not at 24 hrs. 

In lines 90-91 it mentions that A represents whole cell, and B represents phosphatydilcholine. 

This is confusing, since I am not sure whether A represents PUFAs for whole cell and B represents PC for cell membranes.

Whithin the figure both A and B look the same and while reading is not clear what is represented. 

In figure 2 the levels of PLA2 and COX 2 were measured from membrane enriched fractions and from whole cell homogenates. However, it would have been interesting to see a measure of enzyme activity. Since AA increases at 1 hr but returns to control levels at 24, whereas PLA2 and COX 2 are highest at 24 hrs than either 1 hr or control.  Therefore, expression levels do not necessarily correlate with the production of AA which needs to be explained. 

Author Response

Point 1

Figure 1 is a little confusing. In lines 78-80 it mentions that there is a significant increase in PUFAs at 1 hr and not at 24 hrs. In lines 90-91 it mentions that A represents whole cell, and B represents phosphatydilcholine. This is confusing, since I am not sure whether A represents PUFAs for whole cell and B represents PC for cell membranes. Whithin the figure both A and B look the same and while reading is not clear what is represented.

We are grateful to the reviewer for the comments, thus we tried to make the reading more easier than before.We also realized that there was an editorial problem during the upload of the paper so that the figure legends were erroneously inserted in the text. Now, we have moved all the figure legends immediately below the figure, changing the size character from 13px to 12px.

Text was modified at page 2, paragraph 2.1, lanes 76-83, 85.

Text between page 3-4, lanes 102-111 was moved above Table I.

Figure 1 and its legend was modified to make the reading of the image easier. Page 3 lanes 87-95

In figure 2 the levels of PLA2 and COX 2 were measured from membrane enriched fractions and from whole cell homogenates. However, it would have been interesting to see a measure of enzyme activity. Since AA increases at 1 hr but returns to control levels at 24hr, whereas PLA2 and COX 2 are highest at 24 hrs than either 1 hr or control.Therefore, expression levels do not necessarily correlate with the production of AA which needs to be explained.

We thank the reviewer for the suggestion, and we agree that measure of enzyme activity could be interesting. Nevertheless, we hypothesize the activation of PLA2/AA/COX-2 cascade since we observed AA enrichment in PC during the first hour from restoration, followed by a decrease in the successive 24 hours, suggesting a release of this fatty acid from PC by PLA2 activity. In parallel, AA remains still higher than control in the whole cells where it is probably substrate for the inducible COX-2 protein. Accordingly, Nito and colleagues demonstrated in an in vivo model of transient focal cerebral ischemia (tFCI) that protein increment correlates with protein activity till the first day (24hours) from reperfusion.

Moreover, we believed that at the base of the inflammatory response there is a lipid reshaping induced by OGD/ogR that leads to AA movement from neutral lipid, probably cholesterol ester, to PC, and that could represent the novelty of the data.

In the manuscript text was modified at paragraph 2.2, page 4, lanes 119-124; page 5-6, lanes 141-146